# Changes in the Mechanical Properties of Alginate-Gelatin Hydrogels with the Addition of *Pygeum africanum* with Potential Application in Urology

**DOI:** 10.3390/ijms231810324

**Published:** 2022-09-07

**Authors:** Jagoda Kurowiak, Agnieszka Kaczmarek-Pawelska, Agnieszka Mackiewicz, Katarzyna Baldy-Chudzik, Justyna Mazurek-Popczyk, Łukasz Zaręba, Tomasz Klekiel, Romuald Będziński

**Affiliations:** 1Department of Biomedical Engineering, Institute of Material and Biomedical Engineering, Faculty of Mechanical Engineering, University of Zielona Góra, Licealna 9 Street, 65-417 Zielona Góra, Poland; 2Department of Mechanical Engineering and Safety, Institute of Mechanical Engineering, University of Zielona Gora, Licealna 9 Street, 65-417 Zielona Góra, Poland; 3Department of Microbiology and Molecular Biology, Collegium Medicum, University of Zielona Góra, Licealna 9 Street, 65-417 Zielona Góra, Poland; 4Computer Center, University of Zielona Góra, Licealna 9 Street, 65-417 Zielona Góra, Poland

**Keywords:** biodegradable hydrogels, sodium alginate, African plum, pygeum, urethra

## Abstract

New hydrogel materials developed to improve soft tissue healing are an alternative for medical applications, such as tissue regeneration or enhancing the biotolerance effect in the tissue-implant–body fluid system. The biggest advantages of hydrogel materials are the presence of a large amount of water and a polymeric structure that corresponds to the extracellular matrix, which allows to create healing conditions similar to physiological ones. The present work deals with the change in mechanical properties of sodium alginate mixed with gelatin containing *Pygeum africanum*. The work primarily concentrates on the evaluation of the mechanical properties of the hydrogel materials produced by the sol–gel method. The antimicrobial activity of the hydrogels was investigated based on the population growth dynamics of Escherichia coli ATCC 25922 and Staphylococcus aureus ATCC 25923, as well as the degree of degradation after contact with urine using an innovative method with a urine flow simulation stand. On the basis of mechanical tests, it was found that sodium alginate-based hydrogels with gelatin showed weaker mechanical properties than without the additive. In addition, gelatin accelerates the degradation process of the produced hydrogel materials. Antimicrobial studies have shown that the presence of African plum bark extract in the hydrogel enhances the inhibitory effect on Gram-positive and Gram-negative bacteria. The research topic was considered due to the increased demand from patients for medical devices to promote healing of urethral epithelial injuries in order to prevent the formation of urethral strictures.

## 1. Introduction

In the literature, many research results about hydrogels and their great potential in support of tissue regeneration can be found. Hydrogels, since the 1990s, have been the object of intensive research for many scientists. Their crucial advantage is that they are elastic materials with low stiffness, which allows them to be developed for soft tissue healing because they have similar mechanical properties and improve physiological load distribution in the tissue [1,2,3]. Properties of hydrogel materials (e.g., elasticity, ability to store water, biochemically induced degradation, and ability to be drug carriers) make them belong to the group of biocompatible and biodegradable materials with a large spectrum of use in regenerative medicine [4,5,6].

Properties of the hydrogel materials are related to their polymeric structure and composition as well as the presence of other chemical additives. Hydrogel biomaterials can be obtained from natural and synthetic polymers and their mixtures. Nowadays, hydrogels made from natural polymers are popular, and their applications are related to their biological, physicochemical, and mechanical properties [7,8,9,10,11]. Hydrogel structures built from natural polymers are described in the literature as more compatible than those made from synthetic polymers, but some of the research reveals that natural polymers can induce inflammatory response and immunogenicity. Determining whether a material is biocompatible is still an open research question for large group of materials [12,13]. The phenomena occurring in the tissue-material–physiological fluid system, and above all the biochemical composition of the stresses and strains occurring in different tissues and organs, make it much more difficult to determine the real cellular response to an implanted material [14,15]. Hydrogels are used in many areas of the biomedical industry, pharmacy, biosensing, biotechnology, food industry, and agricultural industry [16] because they can bond inorganic chemical components such as calcium phosphorus [7], and drugs [17] and many other biomolecules [18]. Another advantage of natural and synthetic hydrogel materials is that they can be shaped in many forms, such as capsules [19,20], microcapsules [21], scaffolds [22,23,24], sponges [25], and layered structures similar to the blood vessels [26,27]. Forming of these shapes may be carried out using 3D printing, dip-coating, or solution mixing [22,23]. Laboratory research results show that hydrogels have a great potential in damaged cartilage regeneration [28,29,30,31,32], tissue vascularization and vascular grafts [33,34,35,36], skin and wound healing [37,38,39,40,41] and as a support for pancreas in diabetes [42,43,44,45].

Biomechanical characteristics of the material are crucial for its usefulness and effectiveness in the tissue healing process. The relationship between Young’s modulus value of the tissue and stent material should be the same as possible on macro and micro-scale [46,47,48]. In the case of hydrogel materials, it is the level of cross-linking and the way of polymer chain arrangement that determines the mechanical properties, the rate of bioactive substances release, and the hydrogel degradation [40]. Single-component bioactive structures are rarely used in materials supporting tissue regeneration.

The variety of tissues in the human body creates a field for the development of new hydrogel materials with mechanical properties similar to those of the target tissue. The hydrogel materials are usually developed for treatment and regeneration of the pancreas, cartilage, skin, etc., because they have excellent interactions with soft tissues, but there have still been no reports on how hydrogels interact with other soft tissues such as bladder, urethra, nerves, or stomach membrane. The multitude of active substances available on the market, both natural and synthetic, gives a large number of hydrogel–bioactive substance combinations, which has an influence on the mechanical properties increasing stiffness and strength. Thus, to select the right set of material ingredients, very precise requirements should be defined.

In the proper functioning and regeneration of the male urogenital tract, active substances administered orally play an important role. Stents implanted in the urethra are usually covered with antibacterial substances that do not directly support cell regeneration. At present, the trend in the selection of active substances supporting tissue regeneration is dominated by the use of mature resources in the form of herbs or plant extracts [49,50,51]. Among the natural substances supporting the regeneration and proper functioning of the genitourinary tract, there are mainly rye grass pollen *(Secale cereale)*, *Serenoa repens*, commonly known as saw palmetto, and African plum bark *(Pygeum africanum)* [52,53,54]. According to the current reports, the bark of the African plum called Pygeum (*P. Africana*) has a great potential to increase the future production of safe and effective high-quality drugs for the treatment of benign prostate hypertrophy, prostate cancer, diabetes, malaria, chest pain, gastrointestinal conditions, wound healing, and skin infections [52]. Pygeum is a powdered bark of Prunus Africana (also known as *Pygeum africanum*), an evergreen tree that grows in the mountainous regions of Africa. Both the powder and the lipophilic extract are sold on the market under the same name. The literature describes the composition and pharmacology of pygeum. The bark of Prunus Africana contains atranorine, ink acid, beta-sitosterol and its esters, and ferulic acid and its esters [55,56]. These are the compounds that have been said to improve conditions of mild prostate hypertrophy and enlarged prostate. It was found that oral administration of pygeum inactivates the androgen receptor and inhibits the growth of prostate cancer cells [56,57,58,59,60]. Furthermore, *P. africana* has antibacterial properties, and it has been proved that different extracts in organic solvents are able to inhibit the growth of varies of bacterial species [61,62]. The additional antibacterial activity of hydrogels in inhibiting the growth of bacteria such as *Escherichia coli*, which is the main etiological factor of urinary tract infections, including prostatitis [63], or *Staphylococcus aureus*, as etiological factor of surgical site infection [64], would be of particular importance.

The study aimed to research the influence of modifications of hydrogel based on sodium alginate on the mechanical properties, biological activity, and the degradation process in contact with urine. The modified material should have suitable stiffness to transfer loads from the action of muscles on the urethral tube. The need to modify the properties of alginate hydrogels (e.g., mechanical, biological, and degradation process) for their use in the genitourinary system is necessary due to the specific conditions in the urethra. These conditions are variable and depend, inter alia, on possible pathologies within the urethral tissue, urine flow rate, and urine excretion rate, as well as the influence of adjacent muscles, as the authors have broadly described in their earlier work [65,66].

## 2. Results and Discussion

### 2.1. Results of Mechanical Properties Tests

Tube specimens were tested in a radial tensile test to determine Young’s modulus value. These parameters are crucial to determine the potential of the obtained hydrogels to support regenerative processes in the urethra. The obtained Young’s Modulus was used for selecting samples for the next testing. The important criterion is the stiffness of the material. The stiffness of the samples should be comparable or minimally higher in comparison to the urethra tissue. The too-high stiffness of the material might induce excessive deformations in the tissue and, instead of supporting the regeneration process, could damage it leading to its discontinuity [46,65,66,67,68]. Figure 1 shows the Young’s Modulus for the tested samples described according to the adopted nomenclature (Table 4).

For the hydrogels containing only sodium alginate, it was observed (Figure 1a) that there was no linear correlation between increasing alginate concentration and Young’s Module value, which is not consistent with Stevens et al. [69] who studied hydrogels with sodium alginate content from 1 to 4% wt. and determined a linear relationship between the amount of alginate and Young’s Module value. The results obtained confirm the reports of Liling et al. [70] and Barros [14] who demonstrated that the mechanical properties of the hydrogels based on sodium alginate are mainly influenced by the type and concentration of the cross-linking agent and the presence of additives. Moreover, the obtained Young’s Module values for the samples containing sodium alginate alone were compared with those of the urethra (Table 1). The comparative analysis showed that the hydrogels with sodium alginate content of 7% by weight had Young’s Modulus close to the stiffness of the human urethra, and the results for these samples had the lowest standard deviation. In the next stage of the study, the samples of sodium alginate hydrogels were doped with gelatin with about 7% of alginate and 3% of gelatin (A70/30Z) and 3% of alginate and 7% of gelatin. The Young’s Modulus for these samples of sodium alginate with admixture gelatin are shown in Figure 1b and Table 2. The results showed also that the gelatin admixture decreased the stiffness of hydrogels. The obtained results are consistent with the literature reports and show that the addition of gelatin makes the material more flexible [14,71,72,73,74]. Nergini et al. [73] and Karimi et al. [74] examined samples containing only gelatin for which the Young’s Modulus was in the range of several tens of kPa, while the combination of gelatin with sodium alginate increased this value to MPa, which is shown in the presented results and studies by Barros et al. [14].

Figure 1c shows the Young’s Modulus for sodium alginate (PYG/A70) and gelatin doped African plum bark samples. The addition of African plum bark lowered the Young’s modulus and thus made the material more flexible. The lower the value of Young’s modulus, the greater the elasticity of the material. This observation is very important from the point of view of the possibility of using this material to treat urethral disorders, which is an extremely elastic and highly deformable tissue [65,66]. The material to be used in urethra tissue must be sufficiently resistant and at the same time highly deformable. Implantation of a material with too low elasticity (too stiff) into the urethra may prove unsuccessful due to the conditions in the urethra. Such a material, as a result of high resistance to the prevailing stresses and strains in the urethra, may be deformed or crushed. The reduction in Young’s modulus by the addition of African plum bark is considered to be due to the fact that complex molecular compounds, such as atranorin, ink acid, beta-sitosterol and its esters, and ferulic acid and its esters, were introduced into the polymer network during cross-linking, which reduced the interactions and binding forces between bivalent barium cation and G-blocks of sodium alginate, as confirmed by FTIR-IR tests.

### 2.2. FTIR-ATR

Figure 2 shows the FTIR-ATR spectrum for the hydrogel samples of sodium alginate 70 mg/mL cross-linked with 1.5 mol barium chloride (A70) and the samples of 70 mg/mL sodium alginate doped with African plum bark (PYG/A70).

The spectrum shows the characteristic peaks shown in Table 2.

The obtained results are consistent with the outcomes of other research teams [19,82] and confirm that the presence of compounds of complex structures, containing many hydroxyl groups, influences the binding force of G blocks by bivalent cations. The peaks in the range of 3000 to 3600 cm^−1^ for the samples doped with African plum bark are much more intense than for pure alginate, which indicates that hydrogel samples with bark addition contain more compounds with the embedded group -OH, such as esters, alcohols, and acids, which are components of the African plum bark extract. In contrast, the peaks in the 1400–1600 cm^−1^ range, characteristic of bivalent cross-linking cations, are higher in the case of pure alginate compared to the samples with bark addition, which indicates that there are fewer cross-linking cation–barium ion bonds in the samples with bark.

### 2.3. Resorption

The examined hydrogels were subjected to resorption tests realized in the artificial urine. The tests were conducted on a urine flow simulation station. During the tests, a sample placed in a glass tube was subjected to urine infusions during which images were recorded. The research methodology proposed by the other researchers [15,83,84] is based on the changes in the weight of a given sample at the time of the immersion in solution. This assumption was not applied in this study because it does not reflect the real-time of contact between the stent and urine. The immersion tests and weight determination of material loss would not give full information about the behavior of the tubular sample during the urine flow. In the studies described, it was crucial to obtain information on how the hydrogel tube swelled and when it was washed out of the tube by the urine stream. Figure 3 shows examples of photos for sodium alginate samples with 7% weight polymer content and the ones doped with African plum bark and gelatin. The images show the sample in the null state, after a dozen or so infusions, and in the last infusion followed by the displacement of the sample from the system.

Hydrogel samples containing an admixture in the form of African plum bark extract already swelled in 15 infusions, filling the entire volume of the tube, which was not observed for the A70 sample, which was not admixed. This sample was rinsed out only after 161 infusions. The gelatin present as an additive accelerated the resorption and swelling process. For higher content of gelatin in the sample, faster resorption and loss of geometry were observed, and the sample was washed out already in 48 infusions. The accelerated resorption of sodium alginate-based hydrogel material with gelatin is due to the highly hydrophilic nature of gelatin itself. In additional, the presence of amide and carboxyl groups in the structure of gelatin increases the hybridization process in a liquid environment. Water molecules can penetrate freely into the gelatin-added material and damage its structure, a study by Rezaei et al. [85] so confirms.

Relating the obtained results to the literature data is difficult due to the methodology of the study and the fact that the aim was not to determine when the mass of the sample would decrease to zero, but to indicate the moment when the sample lost its properties and was not able to stick to the implantation site. Assuming that a human being urinates 8 times a day, A70 material is in the urinary system for 20 days.

### 2.4. Antibacterial Tests

Sodium alginate hydrogel samples with a 7% weight content of alginate (A70) and with an admixture of African plum bark (PYG/A70) were selected to test their potential action against Gram-negative *Escherichia coli* and Gram-positive *Staphylococcus aureus.* The study of the growth dynamics of the bacterial culture population in the presence of hydrogel samples showed the ability of hydrogels to inhibit the tested strains (Table 3). The strongest effect was observed for hydrogel samples doped with *P. africana* plum bark. The optical density values determined in time intervals showed statistically significant difference in the growth of both tested strains in the presence of hydrogels compared to the control culture; however, a slightly greater difference was observed for *E. coli* than *S. aureus* (Table 3).

Antibacterial activity of alginate hydrogels against *S. aureus* and *E. coli* is confirmed in the literature [86,87,88,89] and is the result of the absence of free carbon in this material. The studies have shown that the presence of African plum bark extract in the hydrogel enhances the inhibitory effect of Gram-positive and Gram-negative bacteria. *P. africana* extracts contain saponins, alkaloids, terpenoids, flavonoids, and tannins, and these substances are responsible for the antimicrobial properties [61,62]. This secondary metabolites of the plants acts by forming complex with extracellular and soluble proteins as well as bacterial cell walls and lipophilic flavonoids may also disrupt microbial membranes [90,91]. Hence, these compounds may be more potent in solution against Gram-negative bacteria such as *E. coli*, which has a thinner cell wall with a predominant plastic layer with an outer membrane. Our research showed a higher sensitivity of *E. coli* to PYG/A70 hydrogel action compared to *S. aureus.*

## 3. Materials and Methods

Sodium alginate is one of the natural components used to produce bioactive and resorbable hydrogel. The alginic acid sodium salt is produced from brown seaweeds (Phaeophyceae) and contains linear copolymers of acids: β-D-mannuronic (M-blocks) and α-L-guluronic (G-blocks) connected by (1,4)-glycoside bond. The G and M-blocks may be arranged in sequence and cross-linking by cations allows to obtain a three-dimensional polymer structure able to bond a large amount of water. Divalent and trivalent cations such as Ca^2+^, Ba^2+^, Mg^2+^, Fe^2+^, and Al^3+^ bond covalently G-blocks of alginate creating a structure called “egg-box” [40,70,92,93]. There is an ongoing discussion about the manner of the cations bond type in the “egg-box” alginate structure. Theoretical research reveals that the covalent character of the bond depends over the metal ion type, and for calcium and magnesium ions, the bond is less covalent and more electrostatic, but some other research described that alginate can be covalently crosslinked by use of di valent ions regardless of the type of element [94,95,96]. Morch et al. claim that the affinity of the cations to the G-blocks decreases in the following order: Pb > Cu > Cd > Ba > Sr > Ca > Co, Ni, Zn > Mn [97]. Alginic acid sodium salt hydrogels have been the object of much research in the regenerative medicine area. Dimatteo et al. described that alginate hydrogel might be a carrier for mesenchymal stem cells and as an injectable gel that could be placed at wound post-injury. This biocompatibility of alginate hydrogels is related to the similarity of their 3D structure to extracellular matrix (ECM) [40]. Matching the stiffness of the scaffold to the stiffness of the surrounding tissues and optimizing the rate of degradation provide optimal conditions for the development of the cells with normal phenotype, which has a positive effect on the healing process. Scaffolds made up of polymer chains can also be functionalized to provide sites for cell bonding or protein adsorption, allowing spatial control over the density of neighboring cells as well as the availability of cytokines and growth factors. The ability to function and interact between the tissues is directly related to the orientation and level of the polymer chains cross-linking [40,98]. The studies by Huang [99] and Yan [100] indicate that calcium ion-cross-linked sodium alginate stimulates the proliferation and differentiation of osteoblasts in vitro. Properties of the alginate hydrogels depend also on the manner of the cation used as a cross-linking agent. Liling et al. [70] demonstrated that the type of ions used for cross-linking affected the mechanical properties of the obtained hydrogel. In their research, the evaluated mechanical properties of the alginate films were cross-linked with solutions containing zinc, calcium, manganese, and aluminum cations at concentrations of 2% (w/v) each. They proved that hydrogel films cross-linked with calcium ions showed the highest tensile strength among the tested samples. They also demonstrated that with increasing concentrations of calcium ions, the tensile strength increases (to 1.5% CaCl_2_) and then decreases (for >1.5% to 5% CaCl_2_). Furthermore, they also reported that the tensile strength increased with increasing the cross-linking time (2, 4, and 6 min), but these differences were not significant (*p* > 0.05). On the contrary, the value of elongation changed, which was higher for cross-linked hydrogel films of up to 2 min and decreased as the cross-linking time increased. In the study, Drenseikiene et al. encapsulated MG-63 cells in a hydrogel based on sodium alginate and gelatin as a biomatrix and studied their survival and response to the presence of cross-linking agents—calcium and barium ions. A slower increase in MG-63 cell viability was identified in samples cross-linked with CaCl_2_ compared to BaCl_2_. This result was due to differences in the stiffness of the gel as cross-linked samples with BaCl_2_ showed higher module flexibility and lower weight loss during the incubation period [101].

The most suitable technique for forming a tubular hydrogel is to immerse the polymer matrix in alginate and crosslink using divalent cations (Figure 4c). This is confirmed by previous studies describing the crosslinking of sodium alginate with calcium cations, as well as an extensive analysis of the literature describing the effect and relationship between alginate concentration and crosslinking agent [86,99,102,103].

Different concentrations of sodium alginate and 1.5 mol barium chloride aqueous solution as a cross-linking solution were examined. The selection of barium ions to cross-linking was chosen to eliminate calcium ions from the structure of the hydrogel. The calcium may promote the formation of possible calcification within the damaged tissue in the area where the hydrogel was to be implanted. Mori et al. reported that the presence of calcium and phosphate ions might impair tissue regeneration, particularly in blood vessels [104].

### 3.1. Materials

In the study, the sodium salt of alginic acid (Sigma-Aldrich, Poznan, Poland) and gelatin (Sigma-Aldrich, Poznan, Poland) were used as a cross-linking agent—barium chloride (Sigma-Aldrich, Poznan, Poland) at 1.5 mol. Resorption tests were carried out in the artificial urine composed of uric acid (416 mM) (Sigma-Aldrich, Poznan, Poland); sodium chloride (154 mM) (Sigma-Aldrich, Poznan, Poland); ammonium chloride (48 mM) (Chempur, Poland); sodium sulfate (34.20 mM) (Avantor Performance Materials Poland S.A.); and sodium dihydrogen phosphate (3.40 mM) (Avantor Performance Materials Poland S.A.) The composition of the artificial urine was selected based on the analysis of the properties of various mixtures performed by Chutipongtanate and Thongboonkerd [105].

### 3.2. Samples Preparation

In the study, the sodium alginate solution was obtained by mixing sodium salt of alginic acid with deionized water for 12 h. The test concentrations of sodium alginate were 70; 80; 90 mg/mL. After mixing in the beaker, the sodium alginate solution was shaken in the ultrasonic scrubber for 15 min to remove air bubbles, the presence of which could interfere with the sample formation by immersion of the polymeric matrix-rod. A similar procedure was adopted for the preparation of alginate/gelatin samples—sodium alginate was not mixed with water, but with a prepared solution of gelatin dissolved in water. Depending on the material configuration (Table 4), gelatin concentrations were 30 and 70 mg/mL. The cross-linking of the material was performed with the immersion method from which the material in the shape of tubes (Figure 4c) was obtained. In this way, the prepared materials of tubular shape were subjected to further experiments: degradation during the urine flow, mechanical tests, and FTIR analysis. Hydrogel samples were formed in the form of tubes by immersion of a polymer rod of 5 mm diameter in the following solutions: cross-linking agent; sodium alginate; cross-linking agent; deionized water (to obtain one hydrogel tube, one complete cycle was performed—the rod was immersed only once in each solution). The forming of hydrogel tubes was carried out with the use of the stand designed and manufactured for the coating application by immersion (Figure 4a). The device was equipped with a control system, thanks to which the fluidity of immersion and emergence was obtained as well as it was possible to adjust the rate and time of the sample immersion. Based on the preliminary tests, the conditions of the tube forming were determined: the rate of immersion and emergence of the matrix 15 mm/s to the depth of 30 mm and the time of immersion maintenance—2 min. These conditions allowed the formation of alginate hydrogel tubes with the following dimensions: inner diameter, 5 mm; outer diameter, 7 mm (±0.06 mm); and length, 30 mm. A 5 mm long sample was cut out of the middle part of the formed tube for mechanical properties tests. As a result of testing the mechanical properties of different material compositions, the optimal alginate concentration that met the mechanical requirements needed to cooperate with the urethra was selected. Correspondingly, each sample was signed by the symbol considering first the content of alginate and gelatin and the presence of African plum bark (Table 4).

Samples containing the extract from the bark of African plum were prepared with the immersion method with the same immersion parameters, but sodium alginate was not mixed with water but with a drained extract from the bark of African plum obtained by infusing with 2 g of bark per 100 mL of deionized water.

### 3.3. Tests of Mechanical Properties

The mechanical properties tests of the hydrogel samples were carried out for all combinations of the samples. Based on the selection and fulfillment of the criterion of similar flexibility to those of the urethra, the samples for subsequent tests were selected. The samples for mechanical properties testing were cut out of the prepared hydrogel tubes of 5 mm length (Figure 4c). The analysis of the mechanical properties of the proposed material was carried out on a testing machine with a Zwick Roel EPZ 005 (Zwick Roell, Ulm, Germany) with the electromechanical actuator. Tube-shaped hydrogel material was subjected to a static tensile test in the radial direction. The test speed was 5 mm/min. Digital photographs were taken in order to accurately measure the geometrical parameters, including the cross-sectional area. The graphical method was used due to the high susceptibility of the samples, which made it impossible to use other methods. The value of Young’s module was determined for the linear range of the relationship stress–strain which was determined individually for each sample. The measurements were carried out in five replicates for each material configuration adopted.

### 3.4. FTIR-ATR Tests

The FTIR-ATR tests were performed on the FTIR multi-band spectrometer (THERMO SCIENTIFIC NICOLET iS50). Based on this investigation any changes inside the material during modification and degradation were observed. Absorption spectra were recorded in the range of 500 to 4000 cm^−1^ using the ART detector with a resolution of 16 scans per spectrum with optical resolution 4 cm^−1^. Before and after each sample measurement, the ATR crystal was thoroughly cleaned with an alcohol wipe. In order to correctly collect the spectra of the tested samples, the background spectrum was measured and collected before each measurement. Tests were conducted under ambient conditions. Before the analysis, the samples were air-dried for 24 h to avoid the effect of “blurring” of the FTIR spectrum shape, shifting of peaks, and changes in their intensity that could be caused by the presence of water. The measurements were carried out in triplicate for the selected materials.

### 3.5. Tests of Antibacterial Properties

Antimicrobial activity of hydrogels was tested by the population growth dynamics of bacterial culture. Activity was tested against two reference strains *Escherichia coli* ATCC 25922 and *Staphylococcus aureus* ATCC 25923. In the procedure, single colonies from the cultures on the agar medium (Graso Biotech) were suspended in the liquid LB medium (Becton Dickinson) to obtain the initial optical density (OD 600 about 0.03).

Discs of hydrogel samples with a 10 mm diameter and 5 mm height, were prepared from a sterile water by using sterile tubes and glass. Discs were placed in one milliliter of strain suspension. Untreated bacterial culture was used as control. One disc from each lot was placed in the medium itself (no bacteria) as a sterility control. The culture was conducted at 37 °C, in an orbital shaker incubator (model ES20 Biosan, Latvia) at 120 rpm, and the dynamic growth was spectrophotometrically monitored at NanoPhotometer NP 60 (Implen, Germany), OD 600 nm. Initially, measurements were taken at hourly and half-hourly intervals for up to 8 h and then after 24 h. The measurements were carried out in triplicates.

Pearson’s chi-squared test was used to calculate the significance of differences in the bacterial growth and the significance level set at *p* < 0.05. The statistical analyses were performed using the program GraphPad (GraphPad Software, USA).

### 3.6. Resorption Tests

Resorption tests of sodium alginate hydrogel samples were conducted in the artificial urine environment. Resorption tests were carried out on a flow simulation stand equipped with recording cameras and a flow control system. This system allowed periodic miction simulation, and as a result, the determined mean quantity of the urine flowed through each sample. Tubular samples were placed in the transparent glass tube that allowed to observe the changes during examination. Adopted research conditions: number of infusions per day: 8 infusions; volume of urine during one micturition/infusion: 70 mL; urine flow rate during micturition: 6 mL/s.

## 4. Conclusions

The paper presents the results of the investigation focused on the modification of sodium alginate hydrogels doped with gelatin and bark of African plum. The research was conducted to develop a material with characteristics suitable for healing urethral damage. As a result of the conducted tests, a material with characteristics similar to those of the urethra was obtained. It is a hydrogel with a sodium alginate content of 7% weight. cured with 1.5 mol barium chloride solution. This material can be doped with African plum bark, whose presence inhibits the growth of Gram-positive and Gram-negative bacteria but affects the elasticity of the material and accelerates its resorption. The presence of gelatin as an additive also makes the material more flexible and accelerates swelling and resorption. The obtained material may serve as a basis for further research on the design of the stent to be implanted in the urethra.

## Figures and Tables

**Figure 1 ijms-23-10324-f001:**
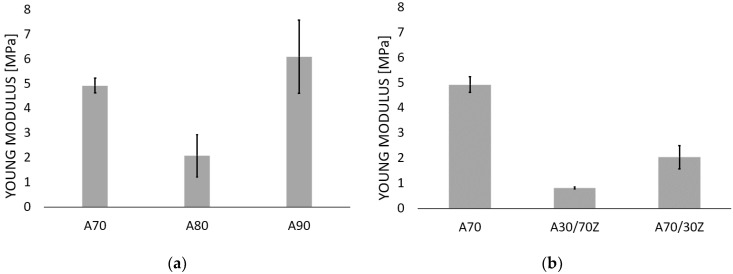
Young’s Modulus values for hydrogel samples consist of: (**a**) sodium alginate, (**b**) sodium alginate mixed with gelatin, (**c**) alginate mixed with gelatin, and the addition of the bark of African plum. All the tested samples were cross-linked with 1.5 mol barium chloride solution.

**Figure 2 ijms-23-10324-f002:**
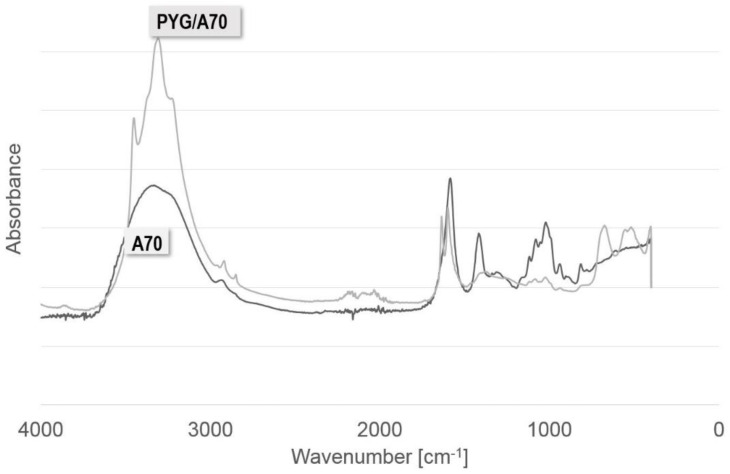
FTIR-ATR spectrum for alginate sodium samples with a concentration of alginate 70 mg. ml cross-linked with 1.5 molar barium chloride (A70) and alginate-Pygeum (PYG/A70).

**Figure 3 ijms-23-10324-f003:**
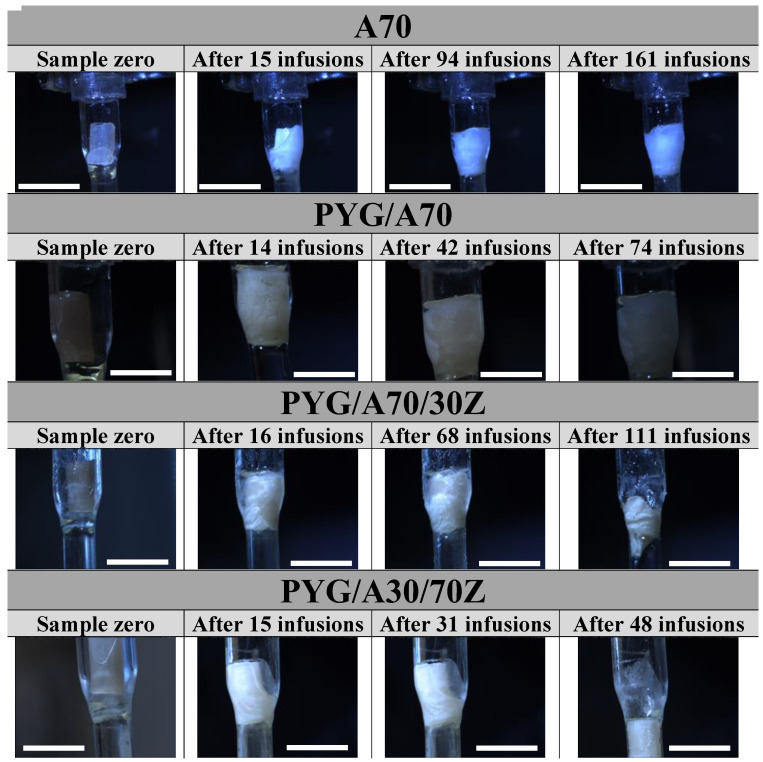
Images from resorption tests during the flow of the artificial urine through the tubular hydrogel samples of sodium alginate 70 mg/mL (A70), sodium alginate with Pygeum (PYG/A70), and sodium alginate–gelatin–Pygeum (PYG/A70/30Z, PYG/A30/Z70). Scale bar: 1 cm.

**Figure 4 ijms-23-10324-f004:**
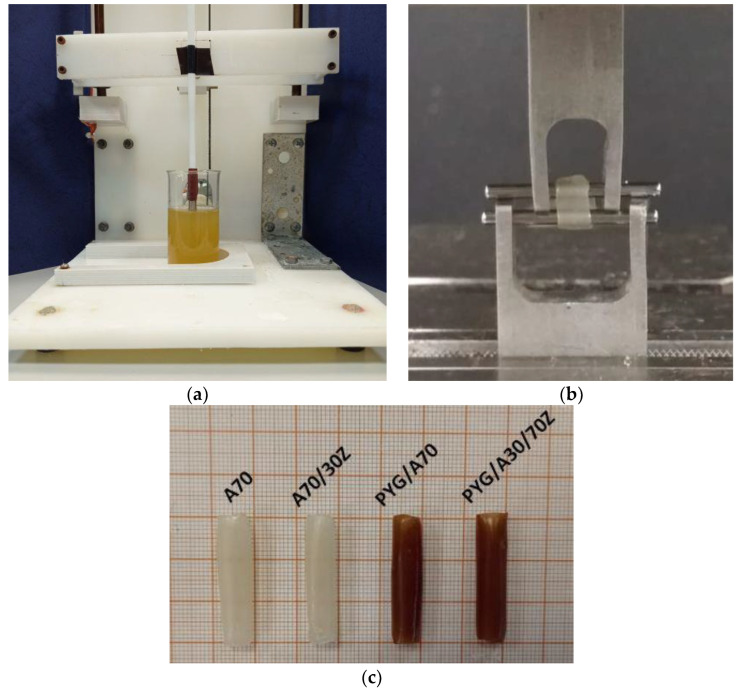
(**a**) Stand for forming hydrogel tubes based on sodium alginate by dip-coating, (**b**) testing the mechanical properties of hydrogel tubes in an axial tensile test, and (**c**) samples in the form of hydrogel tubes obtained by immersion method—examples.

**Table 1 ijms-23-10324-t001:** Summary of Young’s Modulus values of the tested samples about those described in the literature and human urethra.

Sodium Alginate Concentration [wt.%]	Gelatin Concentration [wt.%]	Young’s Modulus	Sample Name
7	0	4.93 MPa	A70
8	0	2.08 MPa	A80
9	0	6.09 MPa	A90
7	3	2.04 MPa	A70/30Z
3	7	0.83 MPa	A30/70Z
7	0	1.47 MPa	PYG/A70
7	3	2.00 MPa	PYG/A70/30Z
3	7	0.36 MPa	PYG/A30/70Z
0	15	45.56 kPa	Nergini et al. [73]
0	25	76.55 kPa	
0	10	70 kPa	Karimi et al. [74]
0	15	80 kPa	
30	65	1000–1200 MPa	Barros et al. [14]
Human urethra		2.4 MPa	Yao et al. [75]
		5 MPa	Spirka et al. [76]

**Table 2 ijms-23-10324-t002:** Characteristic IR peaks obtained for the tested materials.

Wavenumber [cm^−1^]	Assignment	Reference
3000–3600	characteristic of the tensile (valence) bands of the O-H bond	[77,78,79,80,81]
1400 and 1600	characteristic for the asymmetrical and symmetrical tensile vibrations of the carboxylic groups (COO-) and bivalent cross-linking cations	
1000–1200	characteristic of the vibrations of the C-O bond	
800	characteristic of mannuronic acid	
700	characteristic of guluronic acid-binding of crosslinking ions	

**Table 3 ijms-23-10324-t003:** Growth dynamics of *E. coli* and *S. aureus* in presence of alginate sodium samples with a concentration of alginate 70 mg/mL cross-linked with 1.5 molar barium chloride (A70) and alginate-Pygeum (PYG/A70).

Type of Sample	*E. coli* Optical Density Values (OD 600)							
	Time (h)							
	0	1	2	3	4	5	8	24
**PYG/A70**	0.034 ± 0.002 *	0.037 ± 0.002	0.045 ± 0.004	0.066 ± 0.003	0.073 ± 0.002	0.083 ± 0.002	0.787 ± 0.061	2.336 ± 0.066
**A70**	0.032 ± 0.002	0.045 ± 0.002	0.063 ± 0.002	0.084 ± 0.004	0.106 ± 0.005	0.128 ± 0.011	0.767 ± 0.111	2.159 ± 0.089
**Control**	0.032 ± 0.002	0.064 ± 0.005	0.102 ± 0.009	0.130 ± 0.001	0.304 ± 0.027	0.613 ± 0.115	1.232 ± 0.059	3.236 ± 0.028
**Type of Sample**	***S. aureus* Optical Density Values (OD 600)**							
	**Time (h)**							
	0	1	2	3	4	5	8	24
**PYG/A70**	0.034 ± 0.004	0.056 ± 0.006	0.082 ± 0.003	0.117 ± 0.015	0.223 ± 0.02	0.443 ± 0.088	0.781 ± 0.042	3.269 ± 0.171
**A70**	0.034 ± 0.004	0.058 ± 0.009	0.097 ± 0.004	0.141 ± 0.014	0.333 ± 0.028	0.643 ± 0.033	1.187 ± 0.021	2.842 ± 0.045
**Control**	0.034 ± 0.004	0.074 ± 0.005	0.116 ± 0.005	0.171 ± 0.012	0.57 ± 0.054	0.856 ± 0.037	1.366 ± 0.028	4.09 ± 0.065

* Values with standard deviation; statistical significance *p* < 0.05—marked in green; *p* < 0.0001—marked in red.

**Table 4 ijms-23-10324-t004:** Name of the samples and component concentrations in these samples.

Sodium Alginate Concentration [wt.%]	Gelatin Concentration [wt.%]	Content of Pygeum	Sample Name
7	0	-	A70
8	0	-	A80
9	0	-	A90
7	3	-	A70/30Z
3	7	-	A30/70Z
7	0	+	PYG/A70
7	3	+	PYG/A70/30Z
3	7	+	PYG/A30/70Z

## Data Availability

Not applicable.

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
