# Peer review of "Changes in the Mechanical Properties of Alginate-Gelatin Hydrogels with the Addition of Pygeum africanum with Potential Application in Urology"

_ijms, 2022, doi:10.3390/ijms231810324_

Round 1

Reviewer 1 Report

Reviewer report for manuscript IJMS-1843445

The manuscript titled “Enhancing of mechanical properties of Alginate-Gelatin Hydrogels by addition of Pygeum ” aimed to develop hydrogel tubes intended to be used as catheters for the genitourinary tract. The manuscript exhibited promising results regarding the potential of the developed system. However, there is no mention of statistical analysis of the experimental data. Moreover, the manuscript needs to be improved regarding the data presentation, and discussion before acceptance. It is possible to find attached a PDF file containing suggestions for improvement.

Author Response

Dear Reviewer,
    Thank you very much on behalf of myself and the other co-authors of the article "Changes in the mechanical properties of alginate-gelatin hydrogels with the addition of Pygeum Africanum with potential application in urology" for taking the time, reviewing the article and pointing out areas for improvement. The authors have responded to comments by modifying specific sections. 

The responses are attached in the file. 

Reviewer 2 Report

[IJMS] Manuscript ID: ijms-1843445

Entitled: Enhancing of mechanical properties of Alginate-Gelatin Hydrogels by

addition of Pygeum.

In the current manuscript the authors were prepared novel hydrogel materials developed to improve the healing of the soft tissues are alterna- tives for using in medicine like regenerative of tissues or in increasing of biotolerance effect in the tissue-implant-body fluids system. The greatest advantages of hydrogel materials are the presence of a large amount of water and the polymeric structure corresponding to the extracellular matrix, which allows to create the healing conditions similar to the physiological ones. 

After reviewing the current manuscript I found that the this manuscript need deeep revision before to be suitable for publications 

1- Extensive editing of English language and style required

2- the research design should be modified 

3- the result and disscusion parts  must be modified 

4- the methods  must be adequately described 

5- the conclusion part should be support by data 

6- All figurs should be improved to be more clear  

Author Response

(The authors gave the same response as above.)

Reviewer 3 Report

The authors of the present study excellently managed to fabricate alginate/gelatin hydrogels of varying compositions with or without the addition of Pygeum extract. The crosslinked samples were then subsequently analyzed by mechanical tests, FTIR, resorption studies as well as antibacterial assays. Unfortunately, in vitro cytocompatibility/cytotoxicity assays were not included.  Although this study revealed quite interesting results some points should be addressed.

In general, this study is aimed to the use of the described material in the therapy of urethral traumas. Also the effect of urine on the material is analyzed. Maybe this should be somehow also included in the title.

The introduction delivers a basic background hydrogels for tissue regeneration, as well as an the Pygeum component. However, most of the cited literature is already several years old. Maybe the authors should add here more contemporary literature. The intention and aim of the work is presented. However, the writing style of the manuscript is quite difficult to read.

Why did the authors decided to use Pygeum? This could be stated clearer.

The subsequent results section contains results and discussion. Maybe it would be good to name it also results and discussion.

In the beginning of the results part, it would be beneficial if the authors show a picture of a fabricated specimen alongside with a description and the respective dimensions.  

In line 131 the authors mention table 2 but there are only two tables 1 in the manuscript. Please rename. Also the column “Content of Pygeum” in table 1 should be Youngs modulus.

In figure 2 “wavelength” should be “wavenumber”.

In figure 3 the pictures are to dark and the quality is quite bad. Please improve. Also the scalebars are missing.

What was the volume/infusion?

In figure 4: since the measurements were carried out in triplicates – why are the errorbars missing? What was the s.d. of the measurements?

In the adjacent experimental section, the materials and methods are comprehensively described.

In figure 5 scalebars are missing.

Line 293: What was the concentration of the gelatin solution?

The resorption tests could be described in more detail with the respective volume and flow rate.

The final conclusion is in general supported by the results. However, when reading the conclusion the aim of the work is quite clear. Maybe, the authors have to rework the title as well as the introduction to allow the reader to understand the aim of the work directly in the beginning and not only in the conclusions.  

Taken together the underlaying work shows a quite interesting approach and some nice results. However, the authors should rework the manuscript in several ways. Title and introduction should show clearer the aim of the work. The discussion is quite short and should be improved. It would be beneficial to show some in vitro cell culture experiments alongside with a cytotoxicity assay. The authors should also critically check their text and figures for errors, mistakes and misspellings. Also the writing style of the manuscript could be improved since it is sometimes quire hard to read.

Author Response

(The authors gave the same response as above.)

Round 2

Reviewer 3 Report

The authors comprehensively responded to the questions and significantly improved the manuscript.

From my side there are no further changes in the manuscript required.

Thus, it is suitable for the publication in IJMS.